# Not All Data Are Unlearned Equally

**Aravind Krishnan[B]*  Siva Reddy[AC]  Marius Mosbach[A]**
[A]Mila – Quebec AI Institute, McGill University
[B]Saarland University
[C]Canada CIFAR AI Chair
`fistname.lastname@mila.quebec`

## Abstract

Machine unlearning is concerned with the task of removing knowledge learned from particular data points from a trained model. In the context of large language models (LLMs), unlearning has recently received increased attention, particularly for removing knowledge about named entities from models for privacy purposes. While various approaches have been proposed to address the unlearning problem, most existing approaches treat all data points to be unlearned equally, i.e., unlearning that Montreal is a city in Canada is treated exactly the same as unlearning the phone number of the first author of this paper. In this work, we show that this *all data is equal* assumption does not hold for LLM unlearning. We study how the success of unlearning depends on the frequency of the knowledge we want to unlearn in the pre-training data of a model and find that frequency strongly affects unlearning, i.e., more frequent knowledge is harder to unlearn. Additionally, we uncover a misalignment between probability- and generation-based evaluations of unlearning and show that this problem worsens as models become larger. Overall, our experiments highlight the need for better evaluation practices and novel methods for LLM unlearning that take the training data of models into account. We publish the code and datasets for our experiments at: ⓞ https://github.com/McGill-NLP/unequal-unlearning

## 1 Introduction

Machine unlearning deals with the problem of removing specific information or knowledge which was acquired during training from a model (Cao & Yang, 2015; Bourtoule et al., 2021). The motivation for unlearning can be two fold: One one hand there is increasing evidence that LLMs memorize a considerable amount of their training data and are also able to reiterate that data verbatim (Tirumala et al., 2022; Carlini et al., 2023; Huang et al., 2024). This is particularly troubling when LLMs memorize and generate personally identifiable information (PII) or sensitive data (Carlini et al., 2021), creating a demand for technical solutions for removing such data from trained models. On the other hand, the increasing capabilities of LLMs have lead to a growing interest in improving the safety of these models (Ouyang et al.; Bengio et al., 2025). While the standard approach for making models safe right now is post-training via preference optimization (Christiano et al., 2017; Rafailov et al., 2023), unlearning offers an alternative approach to remove, e.g., unwanted and potentially harmful information from LLMs (Jang et al., 2023; Barez et al., 2025).

Existing approaches to unlearning can be broadly categorized into two primary methodologies (we provide a longer discussion in Section 5). While some researchers approach the problem from an optimization perspective, others focus on model editing, which identifies and modifies model components that *store or encode* the targeted data. In this paper, we focus on optimization-based approaches for LLM unlearning and **identify a crucial gap in current approaches**: Existing work in LLM unlearning typically treats all data points designated for unlearning equally. This *"all data is unlearned equally"* assumption disregards potential

---

*Work done during an internship at Mila – Quebec AI Institute.

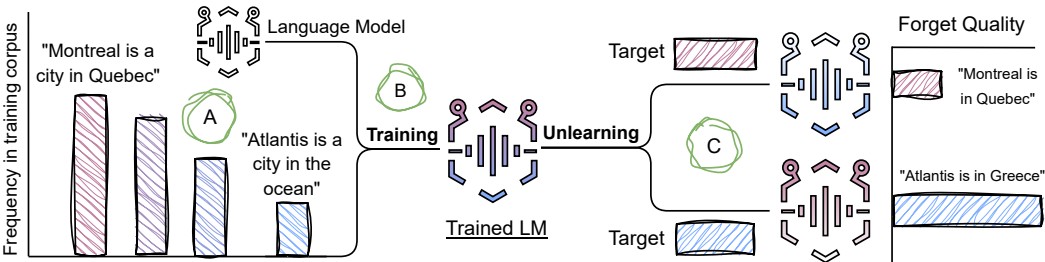

Figure 1: **Illustration of the main hypothesis we explore in this work**. Some knowledge is more frequent in the pre-training data of a model than other knowledge (A). We hypothesize that how well we can unlearn any given knowledge (C) should be strongly influenced by how frequent it is in the pre-training data (B).

variations in data, such as how frequent they are in the training data of a model, and the effect of these variations on unlearning. Hence, we explore the following hypothesis:

*The success of optimization-based unlearning is influenced by the frequency of the targeted information within the training data of a model.*

We hypothesize that frequently seen data are harder to unlearn because the model's "belief" (Hase et al., 2024) about these instances is stronger (cf. Figure 1). To test this hypothesis, we design a comprehensive experimental framework consisting of two distinct settings. First, we conduct a controlled experiment where we train models of increasing size from scratch on fictitious biographies and associated questions. We then unlearn information about frequently and infrequently encountered biographies and compare unlearning performances between these setups. Second, we extended our investigation to a real-world scenario — unlearning factual information from OLMo-7B (Groeneveld et al., 2024) — while controlling for the frequency of the targeted real-world facts.

Our experiments consistently reveal a **strong relationship between frequency in pre-training data and the efficacy of unlearning processes**. We find that methods are more successful in unlearning data instances that are infrequent during pre-training. Highly frequent data on the other hand is either not unlearned at all or only appears to have been unlearned, i.e., it can still be extracted from a model depending on the evaluation method. We also demonstrate that this effect is consistent across different unlearning methods.

In addition to our findings about frequency effects in unlearning, we **highlight potential evaluation issues with unlearning**. Our results show that the efficacy of LLM unlearning varies considerably when evaluated using different methods and that this effect worsens with scale, i.e., larger models are better at retaining seemingly unlearned data in their parameters. We find such disparities also in utility evaluation. Unlearning seems to damage the task-knowledge of a model, e.g., it becomes worse at question answering, while leaving its probabilistic capabilities intact, e.g., it still assigns high probability to seemingly unlearned sequences. Overall, our experiments demonstrate the need for more comprehensive evaluation and the development of data-dependent LLM unlearning approaches.

## 2   Optimization-based unlearning in LLMs

Unlearning can be viewed as a regularized optimization problem that balances two losses: The first loss drives the forget operation and is computed over the target set we wish to forget (**forget set**). The second loss is a regularization term computed over a **retain set**, and aims to preserve model utility as measured on an *independent* **utility set**. This can be formally expressed as:

$$\mathcal{L} = \mathbb{E}_{i \sim forget}\left[\mathcal{L}_{forget}(target_i)\right] + \alpha\mathbb{E}_{j \sim retain}\left[\mathcal{L}_{\text{regularization}}(retain_j)\right] \qquad (1)$$

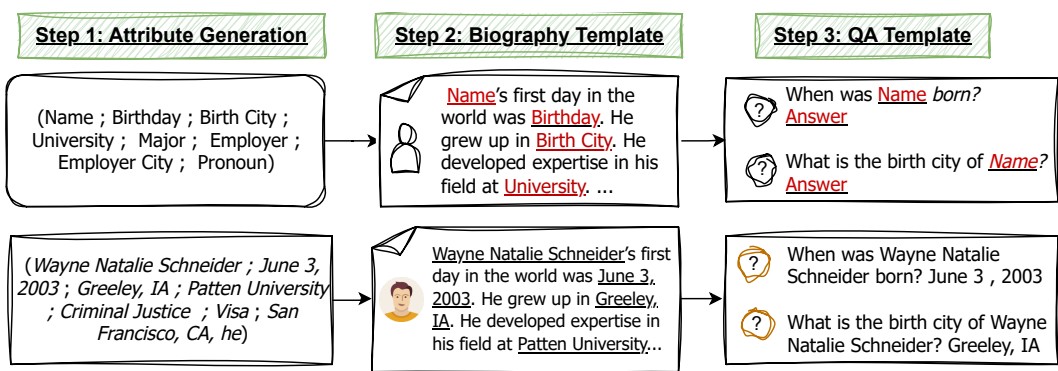

Figure 2: **Data generation process for the pre-training experiments**. (1) we generate attribute tuples randomly. (2) a biography template is chosen to verbalize the biography. (3) QA pairs are constructed using a question template.

Where $\alpha$ is a hyperparameter used to control the strength of the regularization. In previous work, $\mathcal{L}_{forget}$ and $\mathcal{L}_{retain}$ are implemented in different ways (Maini et al., 2024; Yuan et al., 2025; Fan et al., 2024; Yao et al., 2023; Eldan & Russinovich, 2023).

We experiment with three choices of $\mathcal{L}_{forget}$: 1) Gradient Ascent, 2) preference optimization, and 3) refusal training. *Gradient Ascent*[1] is the negated version of cross entropy loss computed over the target set, i.e., given a $(x, y)$ question-answer pair, loss is computed as $\mathcal{L}_{\text{gradient\_ascent}}(\theta, x, y) = -\mathcal{L}_{\text{CE}}(\pi_\theta(y \mid x))$ where $\pi_\theta(y \mid x)$ is the probability that model $\theta$ generates response $y$ when prompted with $x$. For preference optimization, we use the *SIMNPO* loss (Fan et al., 2024), which is a variant of DPO (Rafailov et al., 2023) but without a positive sample. The loss is defined as: $\mathcal{L}_{\text{SIMNPO}} = -\frac{2}{\beta} \log \sigma - \left( \frac{\beta}{|y|} \log \pi_\theta(y|x) - \gamma \right)$ where $\beta$ and $\gamma$ are hyperparameters and $|y|$ is the length of the response. The *refusal training* loss is the cross entropy loss on the sequence "I don't know" across all target samples, i.e., $\mathcal{L}_{\text{idk}}(\theta, x, y) = \mathcal{L}_{\text{CE}}(\pi_\theta(\text{"I don't know"} \mid x))$.

In all cases, we use cross entropy as the regularization loss. Crucially, while the forget loss is computed *only* on the response $\pi_\theta(y \mid x)$, the regularization loss is computed over all tokens in the retain sample to aid unlearning stability (see Appendix A for more details).

## 3 Analyzing frequency effects in a controlled setting

We first construct a synthetic setup which allows us to test our frequency-unlearning hypothesis in a controlled setting. Our setup consists of training language models of various sizes form scratch on a dataset of biographies and questions about these biographies. Once trained, we evaluate the model by asking questions about some of the biographies. We then unlearn information from various biographies that differ according to their frequency in the pre-training data and re-evaluate the models' ability to answer questions about them.

### 3.1 Dataset creation

Here we describe the dataset on which we pre-train our models. We follow the `bioS-single` setup proposed by Allen-Zhu & Li (2024) to create fake biographies. The dataset generation pipeline is shown in Figure 2. Our dataset contains two types of instances — synthetic biographies (BIO ) and question-answer pairs (QA ) about the biographies. We start by sampling a tuple of eight attributes: {`name, birthday, birth city, university, major, employer, employer city, pronoun`}. Next, the attribute tuple is converted into a biography using a randomly sampled biography template. We then convert the attribute tuple into

---

[1]We note even when using Gradient Ascent as an unlearning method, the overall objective is still optimized using gradient descent.

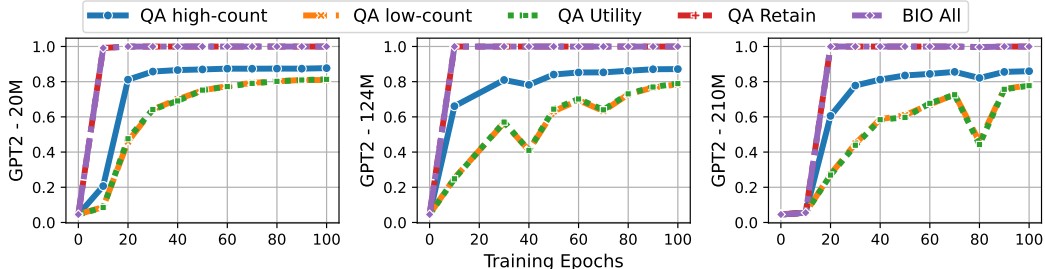

Figure 3: **BIO and QA evaluations as training progresses on the GPT models**. The evaluation metric is Rouge-L. QA accuracies are averaged across all 6 attributes. *Over 100 epochs, the models learn to answer questions from the utility, high and low-count sets which are not shown during training*. High-count biographies are up sampled and so high-count QA is learned faster than the other evaluation sets.

QA instances that query each person-attribute relation (e.g., "When was Natalie Schneider born?", "Where was Natalie Schneider born?"). This is also done using QA templates. We generate 10,000 such biographies and 60,000 associated QA pairs. The attribute lists, biography- and question templates are provided in Appendix B.1.

**Constructing training and evaluation splits**  Before training, we divide the entire dataset (BIO + QA ) into four groups *by person*: retain (50%), high-count (16.7%), low-count (16.7%) and utility (16.6%). To simulate variations in exposure, we up-sample the biographies corresponding to one of the subsets (high-count) by a factor of 10x. We do this by resampling biography templates for this subset, keeping the attribute tuples unchanged. Training is done jointly on BIO and QA instances.[2] For training, we use biographies from *all* splits but the QA instances from *only* the *retain* split. All other QA splits: {*high-count, low-count, utility*} are used for evaluation and unlearning. Note that the QA splits for unlearning are *not* used for pre-training.

Overall, this setting is intended to mimic the acquisition of factual knowledge in LLM pre-training: The BIO instances imitate pre-training data in LLMs and the QA instances mimic instruction finetuning. The goal is to train a model on all biographies but only a subset of the questions, using the other questions to evaluate knowledge extraction abilities.

## 3.2  Training and evaluation

We pre-train four GPT-2 models (Radford et al., 2019) of varying sizes — 20M, 50M, 124M and 210M — on the synthetic data described in the previous section. We follow Allen-Zhu & Li (2024) and randomly sample BIO and QA instances during training.[2]  We pre-train all models for 100 epochs using a learning rate of 0.001. Additional hyperparameters are provided in Appendix B.2.

We evaluate models in two different ways: One tests how well the model retains information about the biographies it saw during training and the other tests the model's ability to answer questions about the biographies. (1) **BIO accuracy**: Here, we test the model's biography completion ability by prompting it with tokens up to a required attribute and then evaluating the completion (eg., "Wayne Natalie Schneider was born on..."). To ensure robustness, we resample biography templates during BIO evaluation. (2) **QA accuracy**: To test a model's general ability to retrieve knowledge stated in the pre-training data, we query the model for person-attribute relations. Since the model also saw QA instances during training, we design this evaluation as a QA task using the {low, high}-count QA , and utility QA subsets, which were held-out during training.

---

[2] Allen-Zhu & Li note that *joint* training leads to better task generalization than pre-training sequentially on BIO instances followed by the QA instances

### 3.2.1 Models are good at answering questions about biographies

Figure 3 shows the evaluation results of the 20M, 124M, and 210M models throughout training. For all models, the BIO accuracy and the QA retain accuracy reaches close to 99% early on. This is not surprising, as the models are explicitly trained on these sets. As training progresses, we see a steady increase in the QA performance for the evaluation splits. For the up-sampled biographies (high-count) we observe a higher performance than for other subsets. However, at the end of training, the accuracies for all evaluation splits are close to each other, plateauing at around 80% accuracy. These results indicates that the trained models are able to perform knowledge extraction for the *unseen* person-attribute questions from the evaluation set, i.e., they learned do extract knowledge from their pre-training data.

## 3.3 Unlearning biography information

Next, we aim to unlearning specific biographical information from our trained models.

**Setup**    Our unlearning setup closely follows Maini et al. (2024). We unlearn only on the QA instances from the targets splits, using the QA instances from the retain set for regularization. Both sets are filtered by the attribute being unlearned. Note that BIO instances are not involved in the unlearning process. We compare Gradient Ascent, SIMNPO and IDK losses for unlearning. We unlearn for 20 epochs using a batch size of 16 and run all experiments with 3 different seeds. Further hyperparameters can be found in Appendix B.3.

**Evaluation**    We evaluate both the QA and BIO accuracies for the target, retain and utility sets. Since the BIO accuracy measures how good a model is at completing information it saw during training via language modeling, we expect this metric to be an upper bound for the QA accuracy. Stated differently, we do not expect a model to be able to correctly answer a question if it cannot correctly complete a biography via language modeling. The metric used both evaluations is Rouge-L (Lin, 2004).

### 3.3.1 Results for individual models

Figure 4 shows the results for unlearning the employer attribute from the 20M model using high-count and low-count groups as unlearning targets. We obtain highly similar results across attributes, model sizes, and unlearning methods, which we discuss in Appendix B.3.

**Unlearning harms knowledge extraction disproportionately**    As unlearning progresses, QA accuracy (top left) drops well below 20% for both high and low count unlearning, showing that the model successfully learns to produce wrong or irrelevant answers for target set questions. This also degrades QA utility (25% drop), suggesting that unlearning adversely affects the model's general knowledge extraction. In contrast, BIO performance for utility sets drops only 10%, indicating that unlearning damages task-specific (QA) knowledge more so than general (biography) knowledge.

**Frequency affects unlearning success**    In Figure 4, the QA accuracies (top left) for both high and low-count unlearning follow similar trends. However, BIO evaluations (top middle) show a stark contrast. Even when QA accuracy drops below 20% by the end of unlearning, the model that unlearns the high-count split retains considerably more BIO accuracy (+20%) than the one unlearning the low-count split. This discrepancy arises from the upsampling of high-count biographies, making it harder to fully erase information from this subset. We see similar trends for SIMNPO as well (Figure 7). Our results for refusal training (Figure 8) are the only exception in this trend, where the BIO accuracy never falls below 98% for either split. We attribute this to the fact that in this setup, the sequence "I dont know" was never seen during pre-training and the model very likely memorizes this response during unlearning.

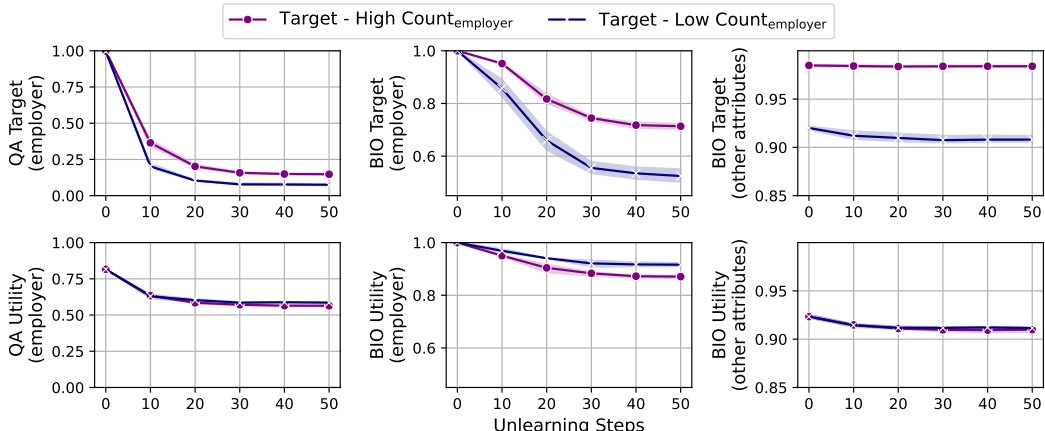

Figure 4: **Target and utility evaluations when unlearning high-count and low count QA splits**. Retain performance never drops below 0.99 during unlearning. All evaluations are Rouge-L scores. Model is GPT-20M, the attribute unlearned is `employer`. *For a comparable drop in target QA performance across both groups (top left), the corresponding biography completion accuracy for the low-count group drops 20% more (top middle).*

### 3.3.2 Scaling trends for unlearning

Next, we investigate how unlearning changes with model size. Figure 5 shows the scaling results for gradient descent. Results for other methods are provided in Appendix B.3. We note that all results discussed are averaged over four attributes and three random seeds.

**Frequency effects persist across scale** Our results show that across different model sizes, the high-count split is consistently unlearned less effectively than the low-count split. This pattern persists also with SIMNPO (see Figure 7), highlighting that pre-training exposure influences unlearning outcomes across both model sizes and unlearning strategies.

**Biography retention increases with scale** When using gradient ascent[3] (Figure 5), we observe a widening divergence between the target `QA` and `BIO` accuracies as scale increases. In other words, larger models are more effective at suppressing the `QA` ability on the unlearning data while preserving the corresponding `BIO` information at the same time. This suggests that as models grow in size, completely erasing information from its parameters becomes increasingly difficult, reinforcing the challenge of *true* unlearning in high-capacity models.

## 4 Analyzing frequency effects in the wild

Finally, we explore if our observations from the previous section extend to even larger models. This helps understand if the implications of our findings extend to real-world scenarios where data sourced from various distributions will need to be unlearned. We choose OLMo-7B[4] (Groeneveld et al., 2024) as the target model, since we have access to its pre-training data. This is crucial for constructing unlearning datasets that differ by frequency. We directly unlearn from the pre-trained OLMo checkpoint.

### 4.1 Experimental setup

**Dataset construction** We curate three real-world QA datasets of the form (source, target, relation), controlling for the relation between source and target: (1) **Country-Capital QA**: These are questions of the form (country, capital, is_capital). For verbalization, we use the

---

[3]Similar trends hold for SIMNPO, see Figure 7
[4]Specifically `allenai/OLMo-7B-0724-SFT-hf`.

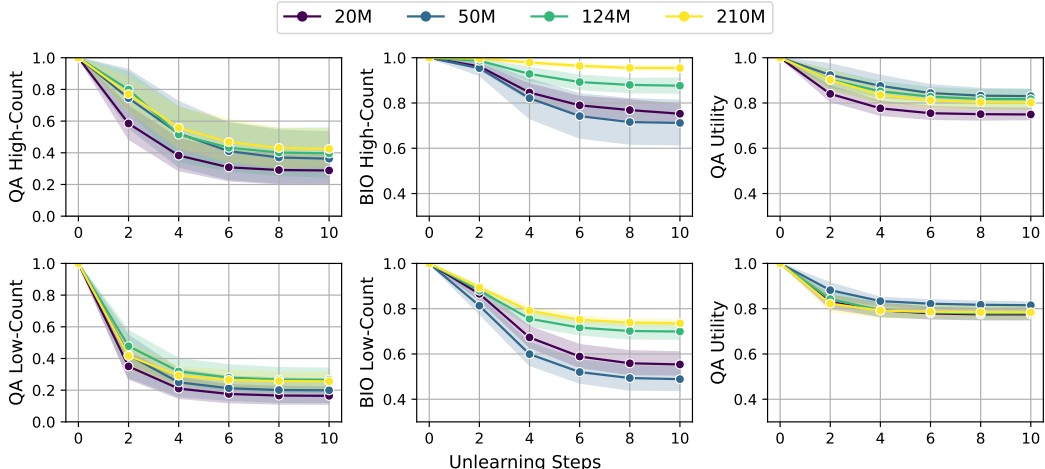

Figure 5: **Unlearning results across GPT sizes** using Gradient Ascent for unlearning. Top row shows results for the high-count split. Bottom row for the low-count split. Results are averaged across four attributes. Rougel-L scores are also normalized with initial values. *Larger models unlearn target QA pairs (left) while achieving higher* BIO *accuracy (middle).*

template "What is the capital of {*country*}? {*capital*}". (2) **Book-Author QA**: Questions of the form (book, author, is_author). Questions are paraphrased in a few different ways using manually created templates. (3) **ZSRE QA**: The ZSRE benchmark contains Wikipedia questions from a variety of (source, target, relation) tuples. This setup represents a loosely controlled albeit realistic dataset for which mimics a potential real-world unlearning request. Samples from each dataset can be seen in Table 4, Table 5 and Table 6.

**Frequency annotation**    Once we collect large samples of (source, target, relation) pairs for each dataset, we compute the *co-occurrence frequency* for each (*source, target*) in Dolma v1.7 (Soldaini et al., 2024), which was used to pre-train OLMo.[5] Co-occurrence frequency is computed as the number of joint appearances of a *source* and *target* pair in Dolma. Let us assume a (source, target) pair (China, Beijing). We wish to estimate the number of times China and Beijing are present in the Dolma corpus, separated by a distance of at most 200 BPE tokens (200 is a conservative upper bound of the word-count in a paragraph). This is formalized as:

$$\text{co-occur}(\text{China}, \text{Beijing}) = \sum_{(i,j)} \mathbf{1}[w_i = \text{"China"} \land w_j = \text{"Beijing"} \land |i - j| < 200] \, ,$$

where $i$ and $j$ range over all token positions in the Dolma corpus. We use the infinigram framework (Liu et al., 2024) to obtain these counts and divide the (*source, target, relation, Dolma count*) data points into three equally sized buckets grouped by *Dolma count*. Table 3 provides an overview of the count statistics. We note that the data at this stage has similarities to our previous setup: We have QA pairs of a fixed *relation*, where the model is *exposed* to one group much more so than the other.

**Unlearning setup**    We use the most frequent 100 data points from the high/low-count buckets for all unlearning experiments. While the count composition of each bucket varies among the datasets, we ensure that the high-count bucket is exposed at-least 100 times more than the low count set for all three datasets.[6] The unearning setup closely follows the synthetic experiments: For each $dataset \in \{capitals, books, zsre\}$, we unlearn high-count$_{dataset}$

---

[5]The SFT model we use was additionally fine-tuned on Tulu (Ivison et al., 2023) but we choose to ignore these counts as they are dominated by the counts in Dolma.

[6]As measured by the median of each bucket.

or low-count$_{dataset}$, using medium-count$_{dataset}$ for regularization. Hyperparameters used across datasets and unlearning methods can be found in Appendix C.2.

The unlearned models are evaluated for unlearning efficacy and utility. Rouge-L is the default metric for evaluation unless stated otherwise. Unlearning efficacy is measured using two approaches: 1) A generative evaluation, where we prompt the model with paraphrased questions from the target set, and 2) a probabilistic evaluation, where the model is prompted to generate a True/False response for a few-shot prompt (Figure 9) that verifies the question-answer pair. We compare the probability of the True/False tokens during probabilistic evaluation. These evaluations are intended to measure unlearning efficacy across task-demands (Hu & Frank, 2024; Deeb & Roger, 2025; Chen et al., 2024).

Utility is also evaluated along two axes: 1) In-domain evaluation: Since the last step in the OLMo-SFT pipeline is instruction-tuning on Tulu V2 (Ivison et al., 2023), we consider the evaluations on Tulu as *in-domain*. We test in-domain generation utility using a zero-shot subset of FLAN (Longpre et al., 2023), which is included in Tulu. We also evaluate on few-shot QA samples from FLAN to assess a *related-but-different* task. To measure probabilistic performance, we compute the perplexity on a subset of Tulu sampled from all composite tasks. 2) OOD evaluation: The world-facts questions from Maini et al. (2024) are used for out-of-domain QA evaluation. We also evaluate on tiny-MMLU and tiny-Hellaswag (Polo et al., 2024), which offer probabilistic measures for utility. To prevent data leakage, we make sure that there is no overlap between the answer tokens of the target and utility splits.

## 4.2   Results

We present results from unlearning the capitals dataset with SIMNPO in Figure 6. Other dataset-method combinations are illustrated in Appendix C.3.

**Unlearning overexposed instances is more difficult**   Mirroring observations from the synthetic setup in Section 3, we see that unlearning efficacy strongly depends on frequency. Across the board, our results show that there is a considerable difference between unlearning data from different count buckets. Given an equivalent drop in utility, we see that the (1) paraphrased target performance for the high-count split does not drop as much as the low-count split (2) in the case that it does, it's probabilistic performance does not drop below 99%. In all our experiments, the generative and probabilistic performance of the low-count split falls consistently lower than the high-count split.

**Probabilistic and generative utility evaluation do not tell the same story**   Across experiments, we consistently observe that all the generation based evaluations drop as unlearning progresses. Notably, zero-shot QA drops more than few-shot QA across the board. On the other hand, neither of our probabilistic evaluations, e.g., tinyMMLU or tinyHellaswag deteriorate with unlearning. This is true even in cases where zero-shot QA deteriorates considerably (see Figures 16 and 17). These results highlight that that there seems to be a task-specific element to utility degradation, which aligns with our findings in the synthetic case (Section 3.3.1). Such task-dependent discrepancies have also shown up in previous work on model evaluation (Hu & Frank, 2024; Hu & Levy, 2023) and preference optimization (Chen et al., 2024). Future work should consider these differences based on evaluation to avoid misconceptions about (decreases in) model utility after gradient-based unlearning.

## 5   Related work

**LLM Unlearning — Methods**   Existing literature on LLM unlearning can be broadly categorized by using either black-box or white-box methods. These categories differ based on whether data-specific modifications are directly applied to the model's parameters or not. Black-box methods rely on gradient-based approaches, i.e., they update model parameters through gradient descent (Ishibashi & Shimodaira, 2023; Jang et al., 2023; Zhang et al., 2024; Fan et al., 2024). White-box methods offer a more surgical approach to unlearning. The general idea is to locate and directly neutralize model components that correspond to the

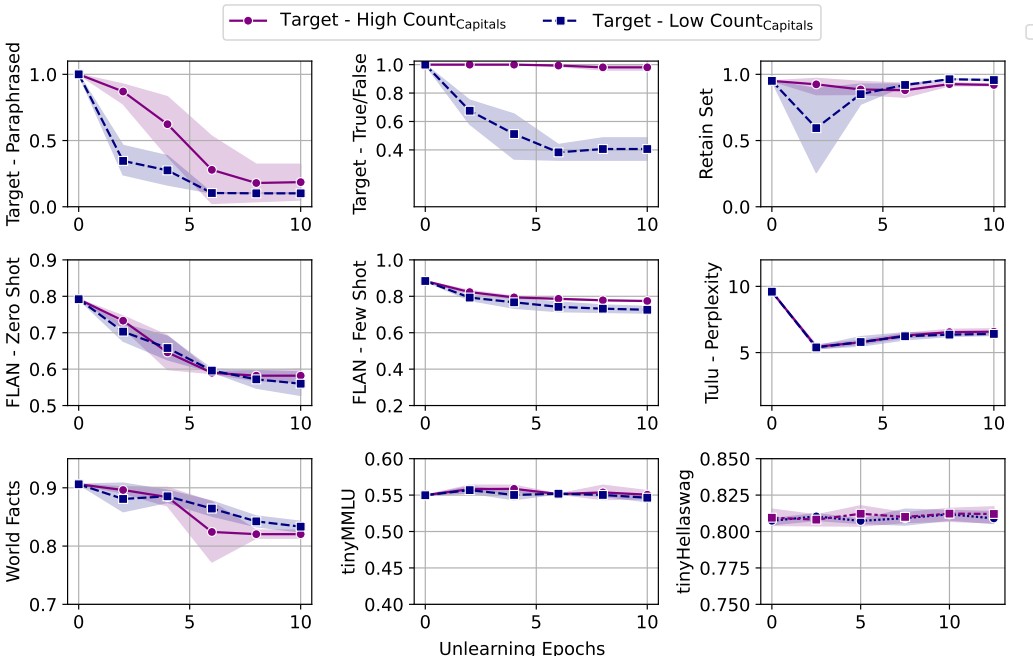

Figure 6: **Unlearning high-count and low-count capitals with SIMNPO on OLMo-7B**. Rouge-L is the evaluation metric. Top row shows performances for the forget and retain sets. Middle row is *in-domain* utility evaluation and bottom row is *out-of-domain* utility evaluation. Target evaluations are normalized across their initial values. *Infrequent instances are forgotten faster and more efficiently across generative (top left) and probabilistic (top middle) evaluations.*

target data, without updating the entire model (Guo et al., 2025; Ilharco et al., 2023; Li et al.). There is an additional class of *grey-box* methods which combine the strengths of the black and white-box methods. These work by limiting gradient descent to a data-specific subset of the model's weights found using attribution (Hong et al., 2025; Zhang et al., 2025) or insights from interpretability research (Tutek et al., 2025).

**LLM Unlearning — Data**   Unlearning datasets are different across tasks and domains. A common task in previous work is to unlearn question-answer pairs and/or multiple-choice questions. Several works attempt to unlearn entire domains, where the data removal is more *general*. Forgetting the Harry Potter universe (Eldan & Russinovich, 2023) or news articles (Shi et al., 2025) are examples for this. Across both tasks, the factuality of the data to be unlearned also varies considerably. Several works look at unlearning fictitious data (Maini et al., 2024; Deeb & Roger, 2025; Guo et al., 2025) from models that are fine tuned on these datasets. Other works also look at removing real-world knowledge like bio-chemical weaponry (Li et al.), Wikipedia knowledge (Patil et al., 2024), or celebrity trivia (Jin et al.) from pre-trained models. Our work spans both setups: we pre-train models from scratch using synthetically generated data, and also unlearn real-world data from a pre-trained LLM.

**Memorization and Unlearning**   The effects of memorization on unlearning have been studied in computer vision (Zhao et al., 2024), where the authors show that verbatim memorization negatively impacts the feasibility of unlearning. Bărbulescu & Triantafillou (2024) study the effects of memorization in LLM unlearning. They unlearn memorized instances of PILE (Gao et al., 2020) from the GPT-Neo models (Black et al., 2021), establishing that verbatim memorization is adversary to LLM unlearning as well. Our work asks a similar question in an intermediate setting: what if instances are not necessarily memorized, but frequently encountered? Does this also correlate with the ability to unlearn data? In concurrent work, Baluta et al. (2024) explore similar questions in a synthetic setup, using machine translation as the target task and gradient ascent as the unlearning method.

Their conclusions agree with ours — that differently exposed/memorized data points are unlearned differently. Our work adds evidence to this narrative, establishing the exposure-unlearning correlation in a widely used open-source language model, i.e., OLMo, and explores variations across model scale, unlearning methods, and evaluation setups.

**Effect of data frequency on LLM behavior**   Variations in data exposure during pre-training has been shown to impact multiple aspects of model behavior: In computer vision, previous work (Parashar et al., 2024; Udandarao et al., 2024) has shown that the zero-shot capabilities of VLMs are linked to the frequency of a concept in the pre-training corpus. Consequently, Verma et al. (2024) show that image generation models are better at imitating images frequently seen during pre-training. In the LLM domain, previous work has shown that the duplication of training samples in the pre-training corpus considerably degrades model performance (Lee et al., 2022) and increases privacy issues (Kandpal et al., 2022). Merullo et al. (2025) show that factual knowledge which is more frequent in pre-training data is linearly encoded in LLM representations. Recent work (Carlini et al., 2023) has also drawn strong connections between memorization and the frequency of training examples: Shi et al. (2024) show that repeated exposure to certain data points during pre-training increases the strength of memorization, making these data points more detectable via membership inference attacks (Mattern et al., 2023). Similarly, Nasr et al. (2025) demonstrate that models trained for more epochs are more likely to regurgitate verbatim training examples, reinforcing the idea that exposure frequency amplifies memorization and affects model behavior at inference time. Our work contributes to this narrative, and studies the role of frequency in unlearning. We examine how repeated co-occurrence of entities in pre-training data can influence unlearning, and show that frequently seen information (here entities and their relationship) is considerably harder to unlearn than infrequent information.

## 6   Conclusions and Future Work

In this work, we test the hypothesis that it should be more difficult to unlearn data that was seen more frequently during pre-training. We provide evidence for this hypothesis across synthetic and real-world setups: 1) We train GPT-2 models from scratch on fake biographies and find that unlearning questions about more frequent biographies is more difficult than other biographies. 2) Extending our results to a real-world scenario, we observe similar trends when unlearning QA pairs from OLMo-7B. Unlearning knowledge that is not frequent in the OLMo pre-training corpus is more easy than frequently encountered data. Additionally, our experiments reveal discrepancies between evaluations of unlearning: While probabilistic evaluations of forget quality and utility display only a minor impact of unlearning on model performance, the impact on performance seems much more negative when using generative evaluations.

We make the following suggestions to guide future research in LLM unlearning: 1) **Unlearning Methodology**: As a first step, future work on optimization based unlearning should group data points in the forget set according to their estimated frequency to make sure that a proposed algorithm works well across different samples. To foster research on data-specific unlearning, we need better benchmarks that offer a range of models where training data is known and unlearning splits can be designed carefully. 2) **Utility Evaluation**: We call for a more comprehensive evaluation of utility following unlearning, given the variation we observed when reporting utility with different benchmarks. We also advocate placing greater emphasis on generative evaluations over likelihood evaluations, since our findings suggest that generative tasks are more sensitive to model degradation.

## Acknowledgments

We would like to thank Fabian David Schmidt for many helpful discussions and feedback on this work. We also thank the COLM 2025 reviewers for their constructive feedback. Marius Mosbach is supported by the Mila P2v5 grant and the Mila-Samsung grant. Siva Reddy is supported by the Canada CIFAR AI Chairs program and the NSERC Discovery Grant program.

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

# A Modified Loss Function for Unlearning

Since our unlearning targets are always QA pairs, and the setup closely resembles Maini et al. (2024) and Fan et al. (2024), we adopt their loss formulation for unlearning. Given a target question-answer pair $(x_{target}, y_{target})$ and a retain pair$(x_{retain}, y_{retain})$, these papers compute the unlearning loss as:

$$\mathcal{L} = \mathcal{L}_{\text{forget}}(\theta, y_{target}|x_{target}) + \alpha\mathcal{L}_{\text{regularization}}(\theta, y_{retain}|x_{retain}) \tag{2}$$

That is, the forget and regularization losses are computed only over the target and retain responses, conditioned on the respective inputs. We propose the use of a modified training loss:

$$\mathcal{L} = \mathcal{L}_{\text{forget}}(\theta, y_{target}|x_{target}) + \alpha\mathcal{L}_{\text{CE}}(\theta, y_{retain}|x_{retain}) + \delta\mathcal{L}_{\text{CE}}(\theta, x_{target}) + \alpha\mathcal{L}_{\text{CE}}(\theta, x_{retain}) \tag{3}$$

The modified function presented in Equation (3) adds additional regularization terms over the question tokens for both forget and retain sets. Keeping the forget loss computation the same, the regularization loss is computed over the target question $x_{target}$, the retain question and the retain answer conditioned on the question. Note that this does not conflict with the forget signal. i.e., we would like the model to forget the answer to a question, but the question itself is a grammatical sentence we can regularize on. We find that the addition of regularization from the question tokens improve the stability of the unlearning process and decrease the drop in QA performance during unlearning, especially for the larger models.

# B GPT-2 Experiments

## B.1 Dataset Creation

**Attributes** Each value in the tuple {`name, birthday, birth city, university, major, employer, employer city, pronoun`} is randomly sampled from the folowing lists:

1. Each name is sampled from a list of first, middle and last names. First names are taken from JTRNS (2025) and last names from CraigH411 (2025). The first name list is reused for middle names, making sure that they are not duplicates.

2. Birthdays are randomly sampled from {[1–28], [Jan–Dec], [1900–2099]}.

3. Birth city is sampled from Grammakov (2025). They are cities in the United States of America.

4. Universities are taken from the list of US universities published by dotJoel (2025).

5. Employer list and locations are sourced from the fortune-500 list compiled by EatMoreOranges (2023)

6. Pronouns are sampled from {he, she, they}. We then use the appropriate personal and possessive pronouns.

**Biography templates and samples** Biography templates are created by first collecting individual templates for each attribute. We use ChatGPT to generate 50 templates *per* attribute. Some example attribute templates are provided in Appendix B.1. Attribute templates are drawn at random and concatenated (in the same order) to make biography templates. This gives us a total number of $50^6$ unique biography templates. This process results in biographies which are six sentence descriptions of each attribute of a person, composed in the same order. Some examples of biographies are shown below:

1. Sandra Denise Wise's arrival happened on February 21, 2053. Her life was first influenced by Mount Croghan, SC. She took advantage of internship opportunities at Hendrix College. She collaborated on research projects in Biology. She was employed at Westlake. She gained industry recognition through work in Houston, TX.

2. `Kathleen Laura Gordon`'s arrival happened on `April 17, 2079`. He was raised in `Talbot, IN`. He spent countless hours in the library at `California Institute of Integral Studies`. He took specialized courses in `Operations Logistics And E-Commerce`. He contributed to the mission of `ConocoPhillips`. He engaged in consulting work in `Houston, TX`.

3. `Christine Margaret Flowers` took their first breath on `June 14, 1900`. Her heritage is rooted in `Falls Mills, VA`. She broadened her academic horizons at `University of Wisconsin--Superior`. She developed programming skills relevant to `Physics`. She achieved professional growth at `Microsoft`. She engaged in consulting work in `Redmond, WA`.

4. `Samantha Megan Deleon` was born on `January 27, 2098`. They entered the world in `Benge, WA`. They refined their analytical skills at `Brazosport College`. They participated in case competitions related to `Human Services And Community Organization`. They developed expertise through `Costco Wholesale`. They developed professional skills in `Issaquah, WA`.

5. `Douglas Scott Kim`'s first day in the world was `June 25, 1902`. He owes his origins to `Wrentham, MA`. He was an active member of the academic community at `Hendrix College`. He learned industry-standard practices in `Music`. He thrived in his career at `JPMorgan Chase`. He advanced his professional journey in `New York, NY`.

**Question Templates**    Question templates are taken from Allen-Zhu & Li (2024). These are:

1. What is the birth date of NAME? BIRTHDAY.

2. What is the birth city of NAME? LOCATION.

3. Which university did NAME study? UNIVERSITY.

4. What major did NAME study? MAJOR.

5. Which company did NAME work for? EMPLOYER.

6. Where did NAME work? EMPLOYER_CITY.

## B.2   Training

**Instance Construction**    We use the mixed training proposed by Allen-Zhu & Li (2024), where the model sees both `BIO` and `QA` instances . Training samples are constructed by concatenating additional samples to each instance, separated by the `<eos>` token. The samples for concatenation are randomly samples and homogeneous to the type of the original instance. That is, QA instances are concatenated together and biographies are likewise concatenated. Samples are concatenated until the tokenized instance has a length of 512. For an dataset of 100 `BIO` and 100 `QA` instances, this produces a dataset of 100x512 `BIO` instances and 100x512 `QA` instances. We use a `BIO` :`QA` token ratio of 1:3 since Allen-Zhu & Li (2024) report that this aids task-learning. This is done by picking more QA instances at random and repeating the instance construction process. For the toy example, the total dataset would contain (100x512 + 300x512) samples for training.

**Training Hyperparameters**    A learning rate of 0.001 is used to train all models. We use a batch size of 32 during training. The models are constructed from scratch; the architectures used can be found in Table 2. The learning rate is warmed up for 10% of total steps. The models are trained with fp16 precision, with a weight decay of 0.01. We use the fused variant of the AdamW optimizer for faster converging. Other training hyperparameters are default values and follow Allen-Zhu & Li (2024). Similar to the original paper, we find that the performances for employer city are comparatively weaker for the trained models, so we do not consider it for unlearning.

| Attribute | Example Templates |
|---|---|
| Birthday | NAME was born on BIRTHDAY
NAME's birthdate is BIRTHDAY
NAME came into the world on BIRTHDAY
NAME was welcomed into life on BIRTHDAY
NAME's journey began on BIRTHDAY |
| Birth City | PERSONAL_PRONOUN was born in LOCATION
LOCATION is where PERSONAL_PRONOUN was born
POSSESSIVE_PRONOUN roots lie in LOCATION
PERSONAL_PRONOUN entered the world in LOCATION
POSSESSIVE_PRONOUN birthplace is LOCATION |
| University | PERSONAL_PRONOUN studied at UNIVERSITY
PERSONAL_PRONOUN enrolled in UNIVERSITY
PERSONAL_PRONOUN was accepted into UNIVERSITY
PERSONAL_PRONOUN completed studies at UNIVERSITY
PERSONAL_PRONOUN honed skills at UNIVERSITY |
| Major | PERSONAL_PRONOUN specialized in MAJOR
PERSONAL_PRONOUN pursued a degree in MAJOR
PERSONAL_PRONOUN studied MAJOR at university
PERSONAL_PRONOUN conducted research in MAJOR
PERSONAL_PRONOUN explored MAJOR coursework |
| Employer | PERSONAL_PRONOUN worked at EMPLOYER
PERSONAL_PRONOUN built a career at EMPLOYER
PERSONAL_PRONOUN gained experience at EMPLOYER
PERSONAL_PRONOUN served in a role at EMPLOYER
PERSONAL_PRONOUN took on responsibilities at EMPLOYER |
| Employer City | PERSONAL_PRONOUN worked in EMPLOYER_CITY
PERSONAL_PRONOUN built a career in EMPLOYER_CITY
PERSONAL_PRONOUN took on responsibilities in EMPLOYER_CITY
PERSONAL_PRONOUN developed skills in EMPLOYER_CITY
PERSONAL_PRONOUN contributed to projects in EMPLOYER_CITY |

Table 1: Example Templates for Biography generation

| Model Size | Number of Hidden Layers | Number of Attention Heads | Hidden Size |
|---|---|---|---|
| 20M | 8 | 8 | 256 |
| 50M | 8 | 8 | 512 |
| 124M | 12 | 12 | 768 |
| 210M | 12 | 16 | 1024 |

Table 2: Model configuration details for the GPT models we train

### B.3 Unlearning

**Hyperparameters** Unlearning uses the loss illustrated in Equation (3). The $\delta$ value is set to 1.0 for all runs. For all unlearning experiments, we search for learning rates among 8e-5, 1e-4, 2e-4, 5e-4 and $\alpha$ values between 1, 5, 10, 15, 20, 25. For the 20M and 50M models, a learning rate of 2e-4 and an $\alpha$ of 20 is seen to work well. The 124M and 210M models use a learning rate of 8e-5 and $\alpha = 25$. For SIMNPO, we use a $\gamma$ value of 0 and a $\beta$ value of 0.1, which is seen to work well.

**Unlearning plots** Figure 7 and Figure 8 show unlearning plots for GPT-2 models using SIMNPO and refusal losses respectively. Results are plotted and averaged across four attributes.

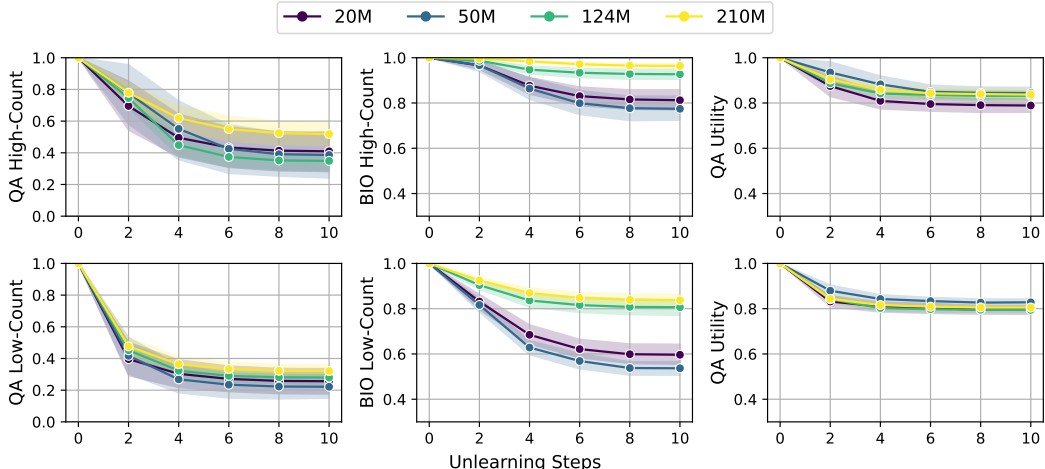

Figure 7: Unlearning GPT models using SIMNPO loss. Results are averaged across four attributes. Top row shows results from unlearning the high-count split and the bottom row from unlearning low-count. The Rougel-L scores are normalized across attributes. *Larger models display an increased ability to seemingly unlearn target QA pairs (left columns), while actually retaining information as seen in target BIO evaluations (middle columns).*

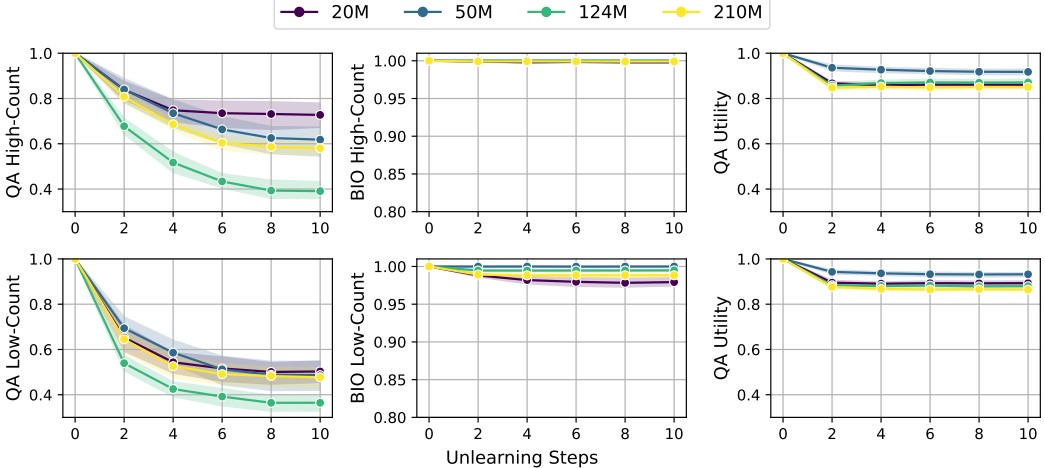

Figure 8: Unlearning GPT models using refusal training. Results are averaged across four attributes. Top row shows results from unlearning the high-count split and the bottom row from unlearning low-count. The Rougel-L scores are normalized across attributes. The models learns to memorize the "I don't know" response that has not been seen during pre-training. This is not seen to affect target BIO performance.

## C  OLMo Unlearning

**Dataset samples and count statistics**  In Table 4, table 5 and table 6, we show the 5 most frequent samples across all three count buckets for Capitals, ZSRE and Books datasets. The associated co-occurence counts in the Dolma dataset (across 200 token spans) are also provided. We also show count statistics for all datasets in Table 3.

| Dataset | Mean ± Std | Median | Min | Max |
|---|---|---|---|---|
| **Capitals** | | | | |
| Low Count | 11,566.42 ± 13,597.12 | 4,897.0 | 1.0 | 44,341.0 |
| Medium Count | 234,492.09 ± 151,210.70 | 202,202.0 | 45,243.0 | 613,776.0 |
| High Count | 3,322,667.0 ± 4,620,916.0 | 1,749,344.0 | 640,168.0 | 27,999,928.0 |
| **ZSRE** | | | | |
| Low Count 100 | 225.5 ± 8.78 | 226.0 | 210.0 | 239.0 |
| Medium Count 100 | 1,950.79 ± 90.37 | 1,939.5 | 1,788.0 | 2,103.0 |
| High Count 100 | 557,792.77 ± 1,324,276.07 | 185,913.0 | 79,823.0 | 9,009,989.0 |
| **Books** | | | | |
| Low Count 100 | 125.99 ± 14.75 | 124.0 | 101.0 | 152.0 |
| Medium Count 100 | 7,526.60 ± 1,214.35 | 7,435.5 | 5,812.0 | 9,991.0 |
| High Count 100 | 32,110.77 ± 29,547.27 | 21,362.0 | 13,255.0 | 209,420.0 |

Table 3: **Frequency statistics for the OLMo datasets**. In all cases, the median of the high count bucket is at-least 100 times more than the low-count group. The median differences are smallest for books, and largest for capitals.

| Low-count [co-occurrence count] | Medium-count [co-occurrence count] | High-count [co-occurrence count] |
|---|---|---|
| What is the capital of **Aruba**? **Oranjestad** [44341.0] | What is the capital of **Slovakia**? **Bratislava** [613776.0] | What is the capital of **China**? **Beijing** [27999929.0] |
| What is the capital of **Marshall Islands**? **Majuro** [41431.0] | What is the capital of **Venezuela**? **Caracas** [563742.0] | What is the capital of **Ireland**? **Dublin** [20868122.0] |
| What is the capital of **Kiribati**? **Tarawa** [41139.0] | What is the capital of **Morocco**? **Rabat** [508836.0] | What is the capital of **France**? **Paris** [18399078.0] |
| What is the capital of **Equatorial Guinea**? **Malabo** [41018.0] | What is the capital of **Colombia**? **Bogotá** [488553.0] | What is the capital of **Israel**? **Jerusalem** [17605351.0] |
| What is the capital of **Cayman Islands**? **George Town** [40836.0] | What is the capital of **Fiji**? **Suva** [488433.0] | What is the capital of **Ukraine**? **Kyiv** [10397531.0] |

Table 4: **Capitals dataset samples:** Samples are shown across count buckets and co-occurrence counts for (**country**, **capital**) pairs in the Dolma corpus for the capitals dataset. *Paraphrased versions are used for testing*

### C.1 Instance construction for unlearning and evlauation

Since the OLMo model is SFT tuned, we provide target and retain instances using the expected chat template. Specifically, the QA instances are constructed as: <|endoftext|><|user|>\nQUESTION\n<|assistant|>\nANSWER where the QUESTION and ANSWER terms are replaced with entries from the target and retain sets. The forget loss is only computed across the answer tokens for the target set. The regularization loss is computed as in Equation (3). The evaluation samples are also provided using the chat template.

### C.2 Hyperparameters

We note that finding appropriate hyperparameters for OLMo was more difficult than the GPT-2 models. Gradient ascent and SIMNPO were both sensitive to hyperparameter configurations. We used the performance of World-facts and the zero-shot subset of flan to monitor model performance, since MMLU and other probabilistic measures did not correlate to zero shot performance. During hyperparameter tuning, we found that model collapse was more frequent than effective forgetting (this has been previously highlighted as a drawback of gradient based unlearning), so good hyperparameter configurations worked well for both target groups. Overall our learning rates were chosen from 2e-6, 3e-6, 5e-6,

| Low-count [co-occurrence count] | Medium-count [co-occurrence count] | High-count [co-occurrence count] |
|---|---|---|
| What company published **Pac-Man Pinball Advance**? **Namco** [239] | What voice type does **Measha Brueggergosman** have? **soprano** [2103] | Who was **Jesus's mother**? **Mary** [9009989] |
| What is the original channel that **Football This Week** played on? **ESPN** [239] | Which show is **T-1000** in? **Terminator 2: Judgment Day** [2102] | The product of **Brewing** is what? **beer** [7161248] |
| What material was used for **Tian Tan Buddha**? **bronze** [239] | When was **Einsteinium** discovered? **1952** [2099] | In what continent is **Canada** in? **North America** [6669104] |
| What is the operating system used with **Kqueue**? **FreeBSD** [238] | What programming language was used to write **DokuWiki**? **PHP** [2095] | Which industry is **MSNBC** associated with? **news** [3059641] |
| What series is the episode **My Sister, My Sitter** part of? **The Simpsons** [238] | What studio produced **Easy A**? **Will Gluck** [2092] | What is the continent that **Mozambique** is located? **Africa** [1521267] |
| What business published **Pro Evolution Soccer 4**? **Konami** [238] | Which state is **Camp Ipperwash** located? **Ontario** [2090] | On what continent can **Libya** be found? **Africa** [1480854] |

Table 5: **ZSRE dataset samples** : Samples are shown across count buckets and co-occurrence counts for (**subject**, **answer**) pairs in the Dolma corpus for the ZSRE dataset. *Paraphrased versions are used for testing*

| Low-count [co-occurrence count] | Medium-count [co-occurrence count] | High-count [co-occurrence count] |
|---|---|---|
| Who is the author of 'Dead to the World'? **Charlaine Harris** [114.0] | Can you name the author of 'Ella Enchanted'? **Gail Carson Levine** [6812.0] | Who is the author behind 'Twilight'? **Stephenie Meyer** [209420.0] |
| Who is the author of 'Deliver Us From Evil'? **David Baldacci** [125.0] | Who is the author of 'Goldfinger'? **Ian Fleming** [9991.0] | Who is the author of the book 'Outlander'? **Diana Gabaldon** [24536.0] |
| Can you tell me the author of 'The Tragedy of Othello, The Moor of Venice'? **William Shakespeare** [128.0] | Who is the author of 'Shiver'? **Maggie Stiefvater** [8032.0] | Who is the author of 'Divergent'? **Veronica Roth** [33856.0] |
| Who is the author of 'The Ruby in the Smoke'? **Philip Pullman** [138.0] | Who is the author of 'The Bone Clocks'? **David Mitchell** [7030.0] | Who is the author of 'The Lorax'? **Dr. Seuss** [17507.0] |
| Who is the author of 'Girl of Nightmares'? **Kendare Blake** [112.0] | Who is the author of 'The Reluctant Fundamentalist'? **Mohsin Hamid** [6353.0] | Who is the author of 'The Secret Garden'? **Frances Hodgson Burnett** [14891.0] |
| Can you tell me who the author of 'Zen and the Art of Motorcycle Maintenance' is? **Robert M. Pirsig** [117.0] | Who is the author of 'The Waves'? **Virginia Woolf** [8272.0] | Who is the author of 'Crash'? **J.G. Ballard** [16081.0] |

Table 6: **Books dataset Samples**: Samples are shown across count buckets and co-occurrence counts for (**book**, **author**) pairs in the Dolma corpus for the Books dataset. *Paraphrased versions are used for testing*

1e-5, $\delta$ and $\alpha$ values of 5,10 worked well. For SIMNPO, $\gamma$ values of 0.1,0 and $\beta$ values between 0.1,1 were searched. Given below are the hyperparameters used for the plots in the paper. Each experimente is run across three seeds.

**Gradient Ascent** For the capitals dataset, we use a learning rate of 3e-6. the $\delta$ and $\alpha$ values are set to 5. For the ZSRE dataset, a learning rate of 2e-6 and $\delta = \alpha = 10$ worked well. For books, we use a learning rate of 2e-6, $\delta = 10$ and $\alpha = 5$.

**SIMNPO** For unlearning capitals, we use a learning rate of 5e-6, $\gamma = 0$, $\beta = 0.1$ and $\delta = \alpha = 5$. For ZSRE, learning rate=5e-6, $\gamma = 0$, $\beta = 0.1$ and $\delta = \alpha = 5$. For books, learning rate=3e-6, $\gamma = 0$, $\beta = 1$, $\delta = 5$ and $\alpha = 10$.

**Refusal training** For capitals, we use learning rate=2e-6 and $\delta = \alpha = 5$. For books and ZSRE, the learning rates are 5e-6, and $\delta = \alpha = 5$.

```
The following are Questions and Answers.  State if the answer to
each question is true or false.

Question: How many planets are in the solar system?
Answer: Eight
True or False: True

Question: What is the tallest mountain in the world?
Answer: Mount Everest
True or False: True

Question: Where was pizza invented?
Answer: France
True or False: False

Question: Who painted the Mona Lisa?
Answer: Leonardo da Vinci
True or False: True

Question: What color is the gemstone ruby?
Answer: Blue
True or False: False

Question: Are sharks mammals?
Answer: No
True or False: False

Question: Can humans breathe underwater without equipment?
Answer: Yes
True or False: False

Question: Who wrote the novel 1984?
Answer: George Orwell
True or False: True

Question: What is the chemical symbol 'Au' for?
Answer: Aluminum
True or False: False

Question: What is the primary ingredient in traditional hummus?
Answer: Chickpeas
True or False: True

Question: What is the capital of China?
Answer: Beijing
True or False:
```

Figure 9: **The true or False prompt used to measure forget quality for OLMo**. For each evaluation, we append the paraphrased question-answer pair from the target set and then measure the compare the model's completion probability for True and False. An example is shown.

## C.3 OLMo unlearning Plots

The following plots (Figure 10, Figure 11, Figure 12, Figure 13, , Figure 14, Figure 15, Figure 16 and Figure 17) show the unlearning trends when learning high and low-count

splits from Capitals, ZSRE and Books dataset using Gradient Descent, SIMNPO and refusal training.

## D    Behind the Scenes

In this section, the authors would like to document the evolution of the paper, all the way from the "plan of a concept" that *AK* and *MM* wrote up to the camera-ready version that the reader sees. Our motivation is to disillusion the linear-progress story that the camera-ready version portrays. The authors thank Benno Krojer for the inspiration.

### D.1    Ideation and Experimental Order

The main idea hatched during a Mont-Royal walk: we'd been thinking about the evaluation of unlearning in general, and found two unconvincing arguments in literature: 1) MMLU results were reported before and after unlearning along with the claim that model utility suffered minimal damage: "What are the boundaries of MMLU utility?" 2) Forget set metrics were averaged: "What does 50% unlearning efficacy on a dataset mean?" . Inspired by Peter Hase's position paper on model beliefs (Hase et al., 2024), we deemed it unlikely that differently strong beliefs can be equally unlearned with one method. When MM was away for EMNLP 2024, AK and MM had a call where OLMo was discussed and the idea of unlearning different buckets of differently exposed data originated. The OLMo part was done first, with a focus on constructing a setup that avoided fine-tuning the model and working on realistic data.

### D.2    Stabilizing OLMo unlearning

Initial unlearning experiments using the loss function from Maini et al. (2024) were very unstable: The standard deviation of the plots was so high that one could not derive conclusions when unlearning different target groups. Initially we thought that the instability arose from the choice in retain set, so we invested a lot of effort in ablating it with not much success. Sometime in January 2025, we tried additional regularization with the question part of the retain set, which stabilized the unlearning trends considerably. The idea to add the forget-question to the regularization came up during a discussion with Fabian David Schmidt two weeks before the submission deadline (which meant that all experiments had to be re-run). While the additional term did not change the OLMo results very much, large GPT-2 models showed considerably less utility hits (comparable to the small models) when it was used.

### D.3    GPT Unlearning

After we observed differences in unlearning and utility estimation for OLMo, we became interested in studying the effect of scale on our trends. MM was looking at Allen-Zhu & Li (2025) and AK started experimenting with Allen-Zhu & Li (2024). Initially we mimicked the TOFU setup. training models on **all** BIO and QA instances, then unlearned a subset of the QA samples. This setup was fragile: trends were quite unstable and we hypothesized that verbatim memorization was working against unlearning. After the QA instances were removed from training (and allowed to be learned through generalization), unlearning started to work well and similar trends to OLMo were observed. Scaling and up-sampling also required some attention: we noticed that if we increased the model size beyond 250M, the model had tendencies to memorize the QA training data and all our bigger models did not generalize well. Scaling biographies to create high-count buckets also faced an overfitting problem: Our experiments with 50x, 100x scaling resulted in overfitting to the high-count bucket and poor generalization.

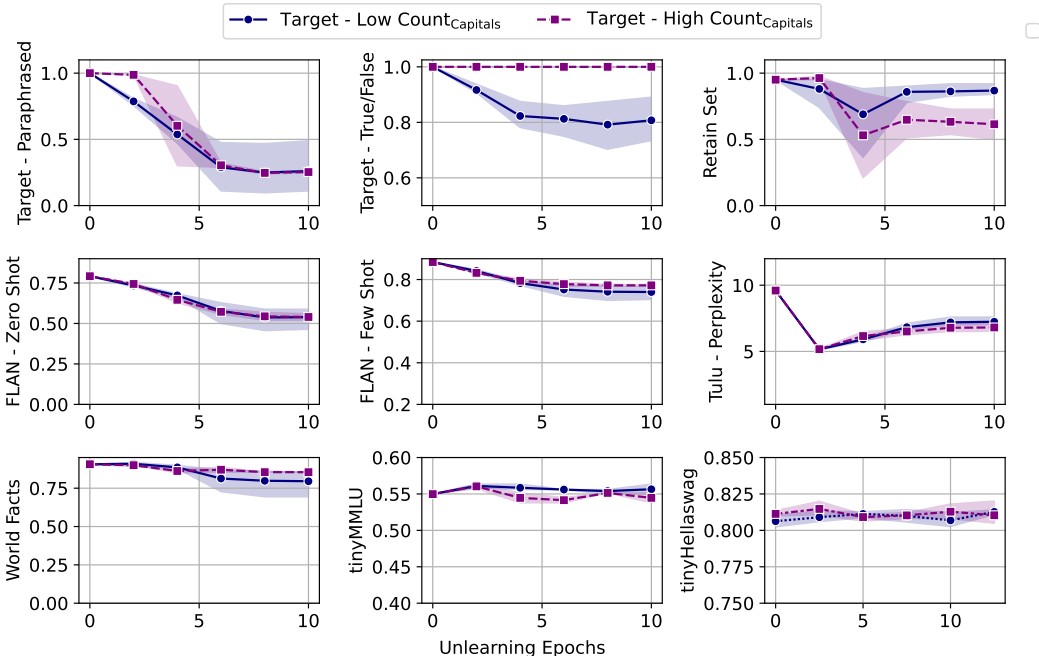

Figure 10: **Unlearning high-count and low-count capitals with Gradient Ascent on OLMo-7B**. Top row shows performances for the forget and retain sets. Middle row is *in-domain* utility evaluation and bottom row is *out-of-domain* utility evaluation. Target evaluations are normalized across their initial values.Low frequent instances are forgotten faster and more efficiently across generative (top left) and probabilistic (top middle) evaluations.

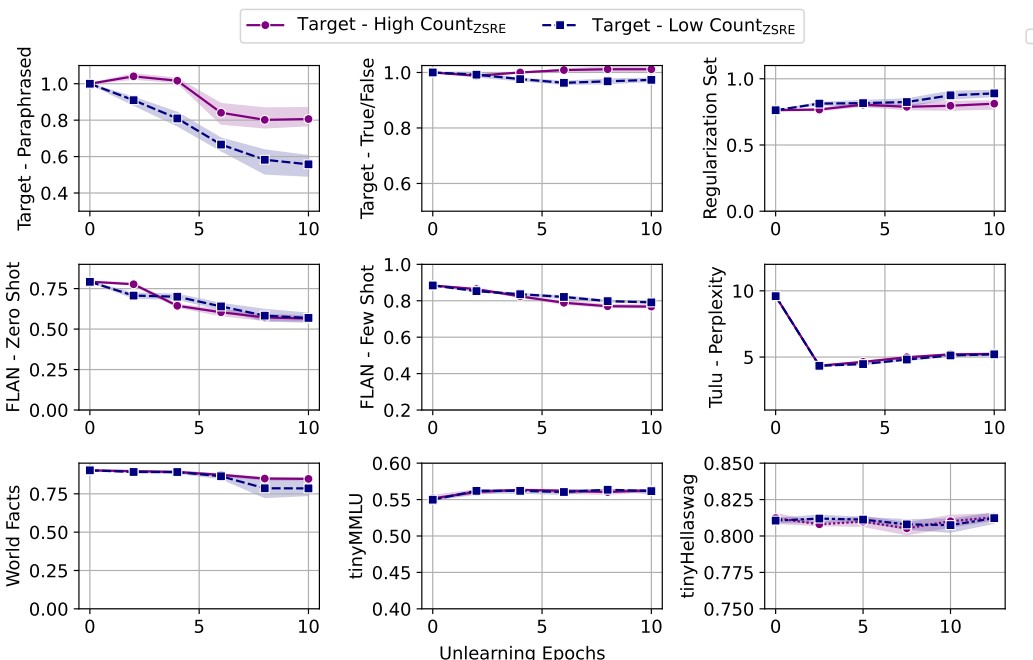

Figure 11: **Unlearning high-count and low-count ZSRE splits with Gradient Ascent on OLMo-7B**. Top row shows performances for the forget and retain sets. Middle row is *in-domain* utility evaluation and bottom row is *out-of-domain* utility evaluation. Target evaluations are normalized across their initial values.Low frequent instances are forgotten faster and more efficiently across generative (top left) and probabilistic (top middle) evaluations.

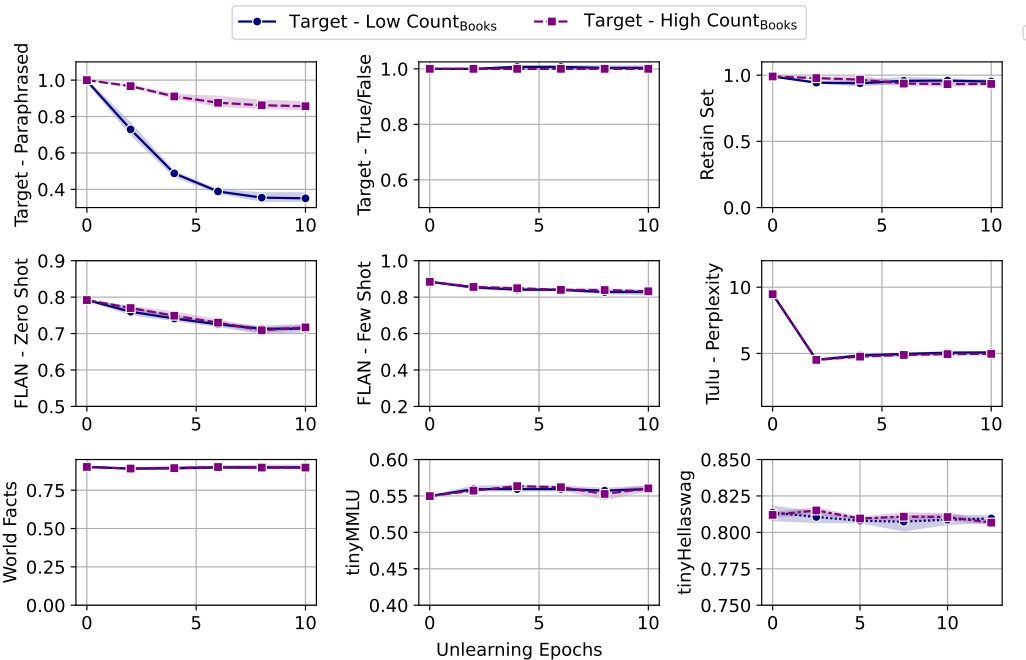

Figure 12: **Unlearning high-count and low-count Books with Gradient Ascent on OLMo-7B**. Top row shows performances for the forget and retain sets. Middle row is *in-domain* utility evaluation and bottom row is *out-of-domain* utility evaluation. Target evaluations are normalized across their initial values. Low frequent instances are forgotten faster and more efficiently across generative (top left) evaluations.

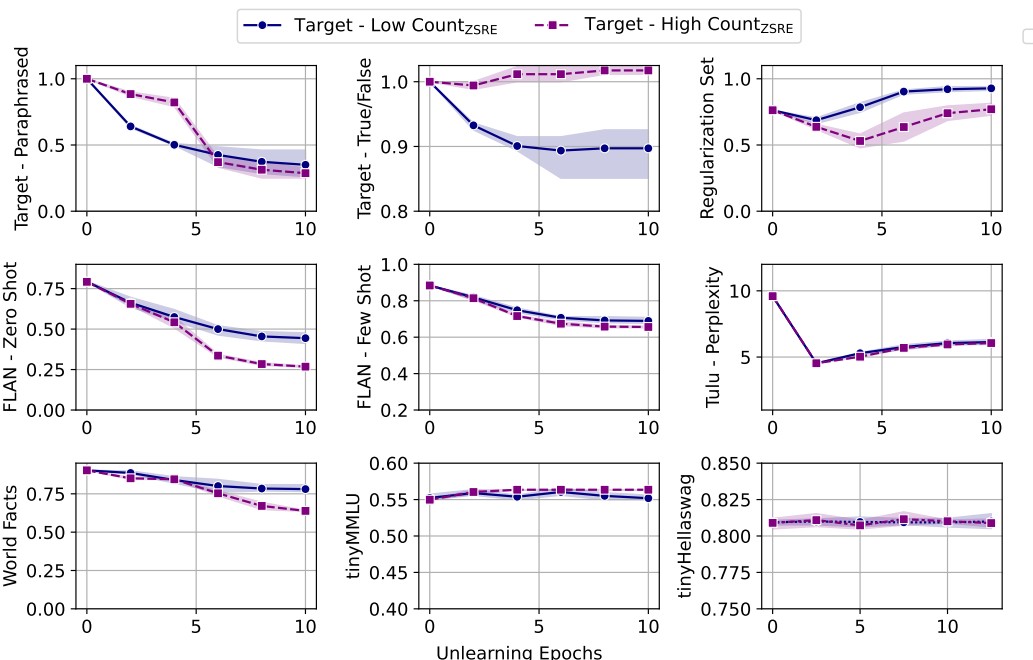

Figure 13: **Unlearning high-count and low-count ZSRE splits with SIMNPO on OLMo-7B**. Top row shows performances for the forget and retain sets. Middle row is *in-domain* utility evaluation and bottom row is *out-of-domain* utility evaluation. Target evaluations are normalized across their initial values. Low frequent instances are forgotten faster and more efficiently across generative (top left) and probabilistic (top middle) evaluations.

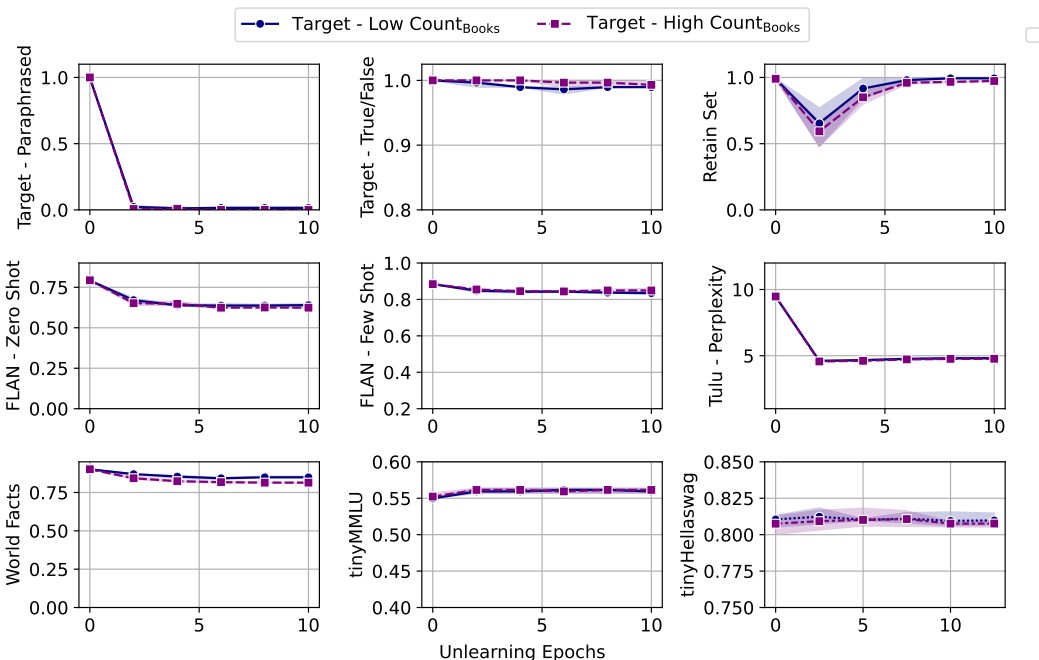

Figure 14: **Unlearning high-count and low-count Books with SIMNPO on OLMo-7B**. Top row shows performances for the forget and retain sets. Middle row is *in-domain* utility evaluation and bottom row is *out-of-domain* utility evaluation. Target evaluations are normalized across their initial values.

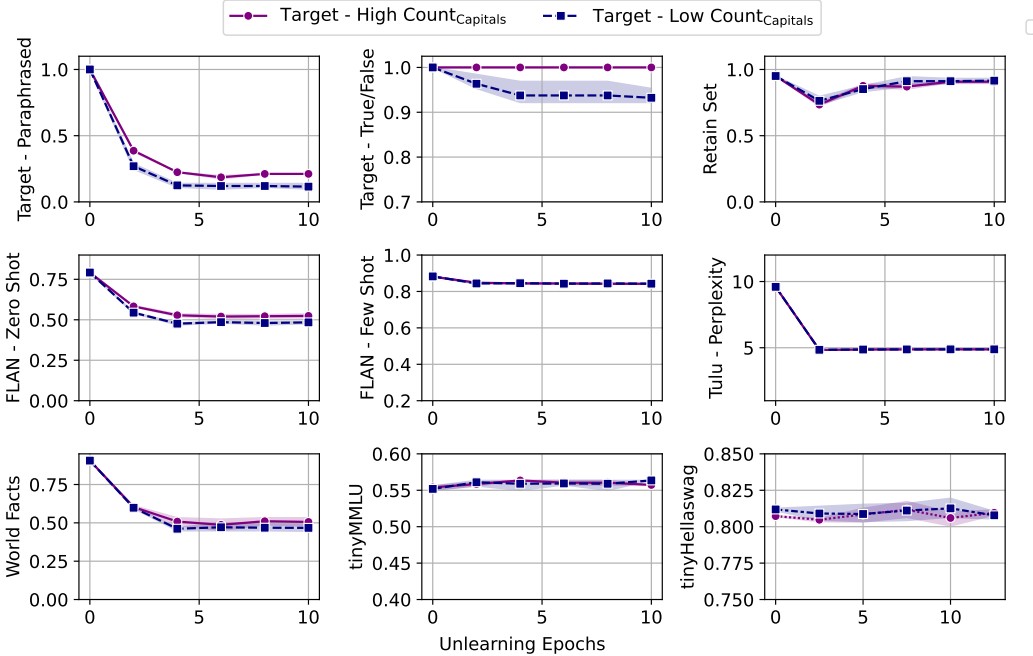

Figure 15: **Unlearning high-count and low-count capitals with refusal training on OLMo-7B**. Top row shows performances for the forget and retain sets. Middle row is *in-domain* utility evaluation and bottom row is *out-of-domain* utility evaluation. Target evaluations are normalized across their initial values. Low frequent instances are forgotten faster and more efficiently across generative (top left) and probabilistic (top middle) evaluations.

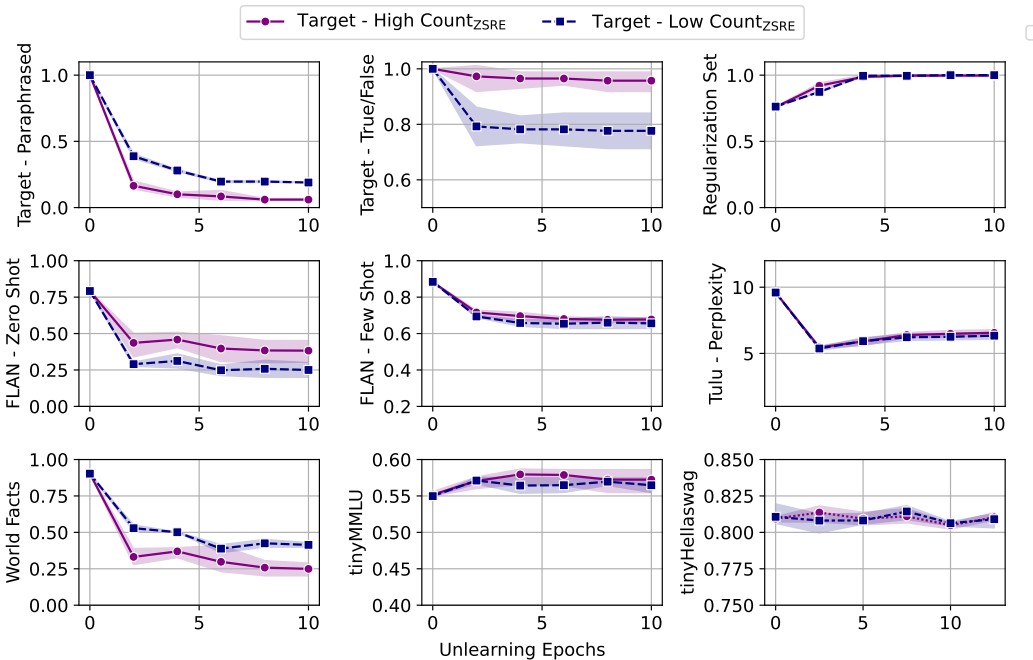

Figure 16: **Unlearning high-count and low-count ZSRE splits with refusal training on OLMo-7B**. Top row shows performances for the forget and retain sets. Middle row is *in-domain* utility evaluation and bottom row is *out-of-domain* utility evaluation. Target evaluations are normalized across their initial values. Low frequent instances are forgotten faster and more efficiently across generative (top left) and probabilistic (top middle) evaluations.

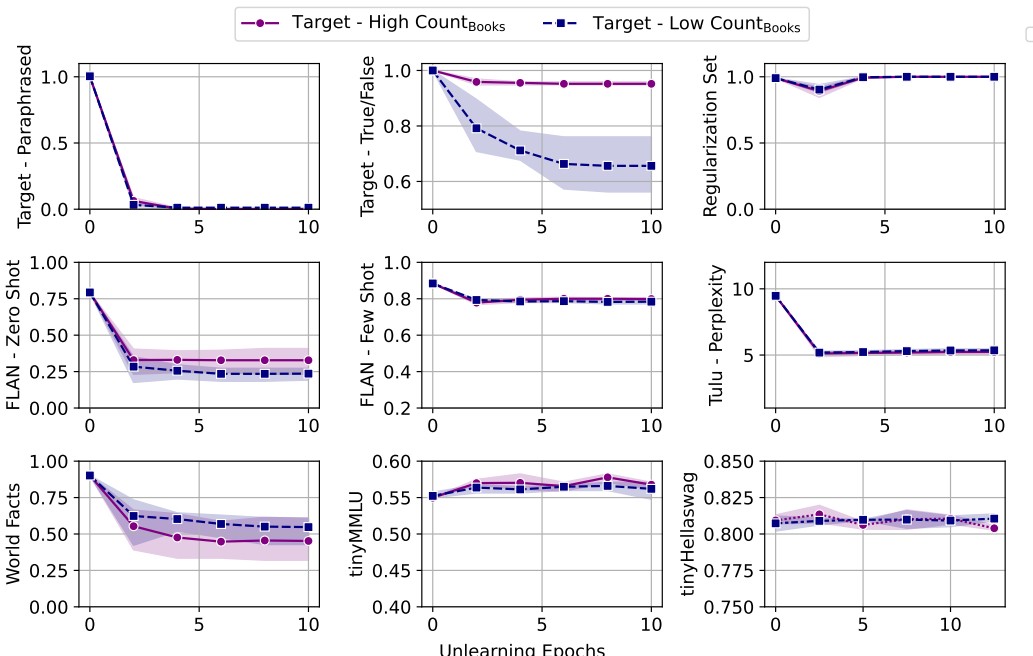

Figure 17: **Unlearning high-count and low-count Books with refusal training on OLMo-7B**. Top row shows performances for the forget and retain sets. Middle row is *in-domain* utility evaluation and bottom row is *out-of-domain* utility evaluation. Target evaluations are normalized across their initial values. Low frequent instances are forgotten more efficiently across probabilistic (top middle) evaluations.

