# OpenReview forum: "Not All Data Are Unlearned Equally"
_colmweb.org/COLM/2025/Conference — COLM 2025_

### Official Review · Reviewer_45nH · 2025-05-11

**Rating:** 6
**Confidence:** 3
**Ethics Flag:** 1

**Summary:**

This paper studies the problem of optimization based approaches for machine unlearning in the context of LLMs. They show that not all data have the same difficulty for unlearning and how more frequent knowledge is harder to unlearn

**Questions To Authors:**

it would be good to add some citations in line 38 (when you refer to existing work that treats all data equally), even if they were cited above.

I wonder if there is a connection to be made with data memorization and membership inference attack methods (https://arxiv.org/pdf/2305.18462), here the idea is to identify data that has been used to train and hence should not be used to evaluate the LLM at inference. In the same line as in this paper, it should be the case that frequency of the data used to train should impact how we evaluate.

**Reasons To Accept:**

- The paper proposes a well done study on machine unlearning techniques for LLMs. It presents prior work issues and shows two set of experiments: (1) a control experiment with small GPT2 models, and (2) an experiment with Olmo as its training data, and thus the frequency of events on it, is available.

**Reasons To Reject:**

- The authors pretrain 4 GPT2 models for the experiments, and all models are in the very small range nowadays (210M params max). While I understand the control experiment, the experiment using Olmo is a lot more reliable. The success of unlearning techniques and the further inference for verification may be very well impacted by the size of the model (i.e., a smaller model may have a lot more difficulty memorizing not frequent training instances). The experiments with Olmo, however, are not detailed enough (200 token span co-occurrence frequency? I would really appreciate more details here as I think it is key for any conclusions after.); the experiment should account for the frequency of its training data not for the data of the controlled experiment. I expect that the curriculum in which the synthetic data (QA pairs) is incorporated into fine-tuning has also an impact on how unlearning can be done in this experiment with Olmo.

Similarly, the dataset generation is not really realistic in real settings; LLMs today are trained on vast amounts of training data and the instances that we may want to unlearn are most of the time examples that are not frequent (e.g., PII instances). Not on controlled examples of QA pairs of data we want to unlearn.

---

> ### Author Response · Authors · 2025-06-01
>
> The authors would like to thank the reviewer for their comments. We would like to highlight and clarify some of the experimental choices to shed some light on the raised concerns, which we believe are already addressed in the paper.
>
> 1. >The experiments with Olmo, however, are not detailed enough...; the experiment should account for the frequency of its training data not for the data of the controlled experiment. I expect that the curriculum in which the synthetic data (QA pairs) is incorporated into fine-tuning has also an impact on how unlearning can be done in this experiment with Olmo.
>
> Like the reviewer notes, the GPT2 experiments offer an extremely controlled synthetic setup: We choose to **train** GPT2 models from scratch here, with biographies and question-answer pairs of different exposure rates. Subsequent unlearning experiments are then compared with the exposure rates, establishing a clear correlation, i.e., higher exposure means lower unlearning efficacy.
>
> The OLMo experiments are separate from the GPT2 experiments. Most importantly, there is **no training or finetuning of any kind in the OLMo setup**, and the target QA sets are entirely different: We take the OLMo model as is, and unlearn different datasets (Capitals/ZSRE/Books) from it, noting the frequency of the data **in OLMo’s training set** (Section 4.1). Our key finding is that the same correlation we found in the synthetic setup also applies here, despite not having trained the model on the forget sets at all. This is encouraging, since we show that **well-defined synthetic setups can closely mimic real-world settings for unlearning**.
>
> 2. > Similarly, the dataset generation is not really realistic in real settings; LLMs today are trained on vast amounts of training data and the instances that we may want to unlearn are most of the time examples that are not frequent (e.g., PII instances). Not on controlled examples of QA pairs of data we want to unlearn.
>
> We agree with the reviewer that most real-world unlearning requests will not be high-frequency instances, and that requests will vary across their exposure rates (PII exposure rates will, for example, vary depending on the history of data available on the person). This is precisely the point of our paper and the reason for our experimental setup. Our contribution is to show that this difference in exposure does influence unlearning performance: The efficacy of unlearning PII requests of high/low histories will be different because the model is differently exposed to these requests.
>
> 3. >The experiments with Olmo, however, are not detailed enough (200 token span co-occurrence frequency?) I would really appreciate more details here as I think it is key for any conclusions after.
>
> We are sorry to hear that the reviewer did not find the OLMo setup clear enough and we will make sure to make it clearer in the camera ready version of the paper. The co-occurrence count refers to the joint appearance of the subject and answer in Dolma (i.e the training set of OLMo). Say a target (subject, answer) pair is (China, Beijing). We wish to estimate the number of times China and Beijing are present in the Dolma corpus, separated by a distance of 200 words (BPE tokens) or less (200 is a conservative upper bound of the word-count in a paragraph). Therefore, we compute
> $$
> \text{co-occur}(\text{China}, \text{Beijing}) = \sum_{(i,j)} \mathbb{1}\left[ w_i = \text{"China"} \land w_j = \text{"Beijing"} \land |i - j| < 200 \right]
> $$
>
> where \(i\) and \(j\) range over all token positions in the Dolma corpus. The counts are obtained using the [infinigram](https://infini-gram.io/) framework. We make sample counts for all OLMo unlearning targets available in Tables 4, 5 and 6, and promise to publish the code with the camera ready version.
>
>
> 4. > it would be good to add some citations in line 38 (when you refer to existing work that treats all data equally), even if they were cited above.
>
> We thank the reviewer for the comment. We will add all relevant citations at this point to present the evidence together, which we agree will facilitate the reader.
>
> 4. > I wonder if there is a connection to be made with data memorization and membership inference attack methods (https://arxiv.org/pdf/2305.18462), here the idea is to identify data that has been
> used to train and hence should not be used to evaluate the LLM at inference.  In the same line as in this paper, it should be the case that frequency of the data used to train should impact how we evaluate.
>
> The authors agree that such a discussion will facilitate the reader by providing additional context for our work. Reviewer `VcPa` also raised this point:  We will include a discussion that covers these topics in the related works section, the first draft of which can be found in our response to reviewer `VcPa`.

---

> > ### Comment · Reviewer_45nH · 2025-06-03
> > **thanks for your response.**
> >
> > Dear authors, thanks for your response. I think the Olmo setup is clearer now. Please make sure to explain this in the camera ready, if accepted. I still think that the evaluation on Olmo should be more central to the paper.
> > Also, besides acknowledging that the papers on memorization and inference attacks need to be cited, I would have liked a bit more discussion on which particular papers you think are important for the conclusions in your paper.
> >
> > I will keep my score, and I encourage the authors to make those changes discussed if the paper is accepted.

---

### Official Review · Reviewer_qvTu · 2025-05-11

**Rating:** 6
**Confidence:** 4
**Ethics Flag:** 1

**Summary:**

This paper examines the common assumption in LLM unlearning that all data points are equally difficult to unlearn. Through empirical analysis, the authors demonstrate that the frequency of the target knowledge in pretraining data significantly influences unlearning difficulty—more frequent knowledge is notably harder to remove. They further uncover a misalignment between probability-based and generation-based unlearning evaluations, a discrepancy that worsens as model size increases.

**Questions To Authors:**

Why are the QA splits designated for unlearning excluded from the pretraining phase? Including these examples during pretraining would ensure that the model actually learns the target knowledge, making it possible to measure the impact of unlearning by comparing the model's behavior before and after the unlearning process.

**Reasons To Accept:**

1. This paper shows that pretraining frequency directly impacts unlearning difficulty for GPT2 of small sizes.

2. This paper highlights a divergence between probability-based and generation-based metrics, which has implications for how unlearning is measured.

**Reasons To Reject:**

1. The QA splits designated for unlearning and utility evaluation should be included during pretraining to ensure that the model has truly learned the target knowledge. Holding them out is unnecessary, as the goal is to measure the change in behavior before and after unlearning. Including them ensures that any observed change is due to the unlearning process rather than a lack of prior learning.

2. For a more practical and comprehensive utility assessment, the evaluation should be conducted on out-of-distribution data—and ideally on tasks other than QA for the systematic analysis part. This better reflects real-world scenarios, where preserving general utility across diverse domains and task types is often more important than task-specific retention.

---

> ### Author Response · Authors · 2025-06-01
>
> We thank the reviewer for their comments. The authors share the concerns of the reviewer, and would like to point to the results in our paper that already shed light on each of the issues raised.
>
> 1. > “Why are the QA splits designated for unlearning excluded from the pretraining phase? Including these examples during pretraining would ensure that the model actually learns the target knowledge, making it possible to measure the impact of unlearning by comparing the model's behavior before and after the unlearning process.”
>
> The BIO evaluations (~ 100% accuracy for all biographies, see Figure 3) already show that the model has learned the target knowledge. However, it is the ability to retrieve it upon request that we additionally wish to evaluate and unlearn with the QA setup. While we can evaluate the effects of unlearning on ‘true’ knowledge via BIO evaluations post unlearning (Figure 4), reliable measures on information extraction cannot be obtained without holding out the target sets.
>
> Another reason to exclude unlearn/utility samples during training is to simulate real world unlearning requests:  In practical use cases, it is safe to assume that a random retrieval request has not been seen verbatim during training, and that the model is relying on its information extraction ability to retrieve the correct answer. Not training the model on target questions simulates this real-world scenario. The same holds for utility tasks, where the standard is to avoid training models on them for reliable evaluations.
>
>
> 2. > For a more practical and comprehensive utility assessment, the evaluation should be conducted on out-of-distribution data—and ideally on tasks other than QA for the systematic analysis part. This better reflects real-world scenarios, where preserving general utility across diverse domains and task types is often more important than task-specific retention.
>
> We’d like to point out that we already performed this OOD evaluation both in the controlled setup (Section 3.3.1) as well for OLMo (Section 4.1). Our findings in both setups show that utility degradations are 1) task-specific and 2) domain-agnostic, i.e., QA unlearning disproportionately degrades QA utility across domains. Below, we restate the main findings from the OOD evaluations (task-specific + task-unrelated evaluations):
>
> - In the synthetic setup, we show evaluations on the utility set (Figure 4), which is OOD with respect to the unlearning distribution given the statistical independence between these sets. We find that QA (task-specific) performance for this set incurs more damage than the BIO (task-unrelated) performance.
>
> -  In the OLMo setup, similar evaluations are made:  We see that zero-shot QA ability (task-specific) degrades across all OOD splits (FLAN, World Facts), while all task-unrelated OOD evaluations like Tulu Perplexity, MMLU, Hellaswag incur minimal deterioration during unlearning.

---

### Official Review · Reviewer_vCPa · 2025-05-12

**Rating:** 7
**Confidence:** 4
**Ethics Flag:** 1

**Summary:**

This paper studies how easy it is to unlearn data that occurs with different frequencies. The authors use fully controlled, synthetic experiments on small models as well as larger experiments at a realistic scale. They identify that facts that occur more frequently in the data are often harder to unlearn. Moreover, they identify an insidious effect whereby larger models are able to suppress the ability to answer questions about certain facts while still being able to express the information at generation time. Another interesting finding is that the generative behavior (measured via QA) is markedly different from the probabilistic behavior (eg MMLU-style questions).

**Reasons To Accept:**

1. The experiments are conducted in sound setups across reasonable scales. Sufficient details are provided for the reader to understand the setting. I think the biography & QA data is a good stage for studying this problem, since it does mirror the factual data we want to unlearn in the real world but it also permits a fine-grained control over the data during training and unlearning.

2. Unlearning is an important problem, and we generally don't know how hard it is to do it in the real world. Insights into the difficulty of this problem for realistic data are extremely valuable.

**Reasons To Reject:**

1. The biggest flaw I can see with the paper is that it barely interacts with the literature that exists on memorization. There are several related phenomena with the success of membership inference [1] and the ability to extract training data from models [2]. I think these ideas are relevant because the frequency of the data in the corpus does play a role in how easily / deeply the model memorizes facts, which drives these related phenomena. Integrating the findings in this paper with the broader narrative would make it much more impactful and useful to people in the field.

[1] Shi et al., Detecting Pretraining Data from LLMs (ICLR 24).

[2] Nasr et al., Scalable Extraction of Training Data from (Production) Language Models (ICLR 25)

2. There is not a clear way forward presented. To be clear, this is not necessary for the paper to be complete, since the analysis of the "unlearnability" of different data is still interesting. However, at least in the conclusion or discussion, it would be good to discuss exactly what the authors think needs to change about unlearning methodology and evaluation practices. It's OK if it's speculation, but it does feel like the paper is a little incomplete without a clear impact from the analysis.

I will raise my score if the above two points are adequately addressed.

Minor:
- I don't think the dichotomy between preference optimization and unlearning makes sense (lines 29-32). These two problems have markedly different goals, though preference optimization algorithms can be repurposed for unlearning.
- There are a few papers on the gap between likelihood and generations causing issues during preference learning (eg [3]). You may want to cite those as additional evidence, because the current point about probabilistic vs generative behaviors is not supported by a lot of evidence.

[3] Chen et al., Preference Learning Algorithms Do Not Learn Preference Rankings (NeurIPS 24)

---

> ### Author Response · Authors · 2025-06-01
>
> We thank the reviewer for their thoughtful comments and we are happy to see that they appreciate our controlled experimental setup and the valuable insights from our experiments.
>
> > The biggest flaw I can see with the paper is that it barely interacts with the literature that exists on memorization. There are several related phenomena with the success of membership inference [1] and the ability to extract training data from models [2]. I think these ideas are relevant because the frequency of the data in the corpus does play a role in how easily / deeply the model memorizes facts, which drives these related phenomena. Integrating the findings in this paper with the broader narrative would make it much more impactful and useful to people in the field
>
> We thank the reviewer for this suggestion. We found the papers recommended very relevant to the discourse and agree that readers will find such a discussion useful. Hence, we will extend the related work section to include a discussion about the role of frequency for these related phenomena.
>
> # Frequency and Memorization
> Variations in data exposure during pre-training has been shown to significantly impact multiple aspects of LLM behavior: Previous work has shown that the duplication of random training samples in the pretraining corpus considerably degrades model performance ([Lee et.al, 2022](https://arxiv.org/pdf/2107.06499)) and increases privacy issues ([Kandpal et.,al 2023](http://arxiv.org/abs/2202.06539)).  Recent work ([Carlini et.al., 2023](http://arxiv.org/abs/2202.07646)) has also drawn strong connections between memorization and the frequency of training examples: [Shi et.al., 2024](https://arxiv.org/pdf/2310.16789) show that repeated exposure to certain data points during pretraining increases the strength of memorization, making these data points more detectable via membership inference attacks ([Mattern et.al., 2023](https://arxiv.org/pdf/2305.18462)). Similarly, [Nasr et.al., 2025](https://arxiv.org/pdf/2311.17035) demonstrate that models trained for more epochs are more likely to regurgitate verbatim training examples, reinforcing the idea that exposure frequency amplifies memorization and affects model behavior at inference time. Together, these works establish a clear pattern: increased exposure leads to stronger memorization, which makes it easier to perform MIA attacks and extract memorized data verbatim. Our work contributes to this narrative, and studies the role of frequency in unlearning. We examine how repeated co-occurrence of entities in pretraining data can influence unlearning, and show that frequently seen information (here entities and their relationship) is considerably harder to unlearn than infrequent information.
>
> > There is not a clear way forward presented. To be clear, this is not necessary for the paper to be complete, since the analysis of the "unlearnability" of different data is still interesting. However, at least in the conclusion or discussion, it would be good to discuss exactly what the authors think needs to change about unlearning methodology and evaluation practices. It's OK if it's speculation, but it does feel like the paper is a little incomplete without a clear impact from the analysis.
>
> We thank the reviewer for the suggestions and we will add the following to the Conclusion section of our paper:
>
> **Unlearning Methodology**: As a first step, future work on optimization based unlearning should group data points in the forget set according to their estimated frequency to make sure that a proposed algorithm works well across different samples. To foster research on data-specific unlearning, we need better benchmarks that offer a range of models where training data is known and unlearning splits can be designed carefully.
>
> **Utility Evaluation**: We call for a more comprehensive evaluation of utility following unlearning, given the variation we observed when reporting utility with different benchmarks. We also advocate placing greater emphasis on generative evaluations over likelihood evaluations, since our findings suggest that generative tasks are more sensitive to model degradation.
>
>
> > There are a few papers on the gap between likelihood and generations causing issues during preference learning (eg [3]). You may want to cite those as additional evidence, because the current point about probabilistic vs generative behaviors is not supported by a lot of evidence.
>
> The authors thank the reviewer for the citation. We will include this citation along with [Hu et.al., 2024](https://openreview.net/forum?id=U5BUzSn4tD#discussion) to illustrate the disparity already observed between these evaluations.

---

> > ### Comment · Reviewer_vCPa · 2025-06-04
> >
> > I am satisfied with the authors' acknowledgment of the relevant works that I raised in my review. The drafted additional text also looks good. I will raise my score to a 7.

---

### Official Review · Reviewer_rsSN · 2025-05-19

**Rating:** 8
**Confidence:** 4
**Ethics Flag:** 1

**Summary:**

This research investigates the relationship between the unlearnability of certain knowledge and its frequency of exposure during the training stage. Through extensive analysis, the authors reveal that knowledge encountered more frequently is harder to unlearn.

**Reasons To Accept:**

- This study offers meaningful insights to the research community on the topic of unlearning, backed by extensive experimental evidence.

- The authors present both a controlled experiment using a synthetic dataset and an analysis in a real-world setting (i.e., analysis on the training data for OLMo), offering a concrete and well-rounded evaluation

**Reasons To Reject:**

For the controlled experiment with synthetic data, it would be beneficial if the authors included results on a mixed dataset (e.g., synthetic BIO data combined with general text) to better assess the generalizability of the findings.

---

> ### Author Response · Authors · 2025-06-01
>
> We thank the reviewer for their positive evaluation of our work and for highlighting that our study offers meaningful insights backed by extensive experimental evidence.
>
> **On the generalizability of our findings**: We’d like to point out that our OLMo experiments already address this point. OLMo was pre-trained on a mixed dataset (Dolma) and it is fair to assume that information we unlearn there (Capitals/ZSRE/Books) is only available in a subset of the data. Yet, we find very similar trends for OLMo compared to our synthetic setup, demonstrating the generalizability of our findings.

---

### Author Response · Authors · 2025-06-10

With the discussion period closing, the authors would like to thank the reviewers for highlighting that our work ` "offers meaningful insights to the research community on the topic of unlearning, backed by extensive experimental evidence" (rsSN)` and that it is `"a well-done study on machine unlearning techniques for LLMs" (45nH).`

We are glad that our carefully designed experimental setup was acknowledged for its rigor and thoroughness: `"The authors present both a controlled experiment using a synthetic dataset and an analysis in a real-world setting, offering a concrete and well-rounded evaluation" (rsSN)`. `"The biography & QA data are a good stage for studying this problem, as they mirror the factual data we wish to unlearn while permitting fine-grained control during training and unlearning" (vCPa)`.

We were also happy to see that the reviewers deemed our findings on unlearning evaluation significant: `"The paper highlights a divergence between probability-based and generation-based metrics, which has implications for how unlearning is measured" (qvTu)`. `"...interesting finding is that the generative behavior (measured via QA) is markedly different from the probabilistic behavior (e.g., MMLU-style questions)." (vCPa)`. The authors agree with the reviewer `(vCPa)` that `"Unlearning is an important problem, and we still lack a clear sense of its real-world difficulty. Insights into this challenge for realistic data are therefore extremely valuable"`.

The reviewers also provided constructive suggestions that we found valuable. Reviewers `vCPa` and `45nH` suggested adding a discussion on memorization and model-inversion attacks. We addressed this suggestion by drafting such a discussion, to which reviewer `vCPa` responded: `"I am satisfied with the authors' acknowledgment of the relevant works that I raised in my review. The drafted additional text also looks good"`. We will include this material in the camera-ready version.

Reviewer `45nH` asked for a clearer explanation of how we compute co-occurrence counts. We offered a clearer explanation of this computation with an example, after which `45nH` replied: `"I think the Olmo setup is clearer now. Please make sure to explain this in the camera-ready, if accepted"`. We will incorporate this clarification as well.

Reviewer `qvTu` expressed doubts about excluding target QA sets when (pre-)training synthetic models. We clarified this experimental choice by pointing out that we cannot obtain reliable measures of information extraction without withholding these sets. Additionally, not training models on the target-sets mimics the real-world unlearning scenario we already test with OLMo. The reviewer did unfortunately not engage in further discussion; but we hope that the clarification addressed the reviewer's concern.

We thank the reviewers again for their feedback and believe the planned revisions will further strengthen the paper.

---

### Decision · Program_Chairs · 2025-07-08

**Decision:**

Accept

**Comment:**

This paper makes a valuable and well-supported contribution by showing that data frequency significantly affects unlearning performance in LLMs. The experiments span both synthetic and real-world settings, and are generally well-executed. While some reviewers expressed concerns about the realism of the synthetic setup and the limited scope of utility evaluations (primarily QA-focused), I view these as natural limitations of scope rather than flaws that undermine the core contributions.

A second concern, around insufficient engagement with related work on memorization and membership inference, was adequately addressed in the rebuttal, with relevant citations and discussion to be added in the camera-ready version.

Given the novelty of the findings, the clarity of the experimental evidence, and the thoughtful response to feedback, I believe the paper merits inclusion.